



**Size-resolved exposure risk of persistent free radicals (PFRs)**
**in atmospheric aerosols and their potential sources**
Qingcai Chen,[a,] Haoyao Sun,[a,] Wenhuai Song,[b] Fang Cao,[b] Chongguo Tian,[c] Yan-Lin
Zhang[b*]
[a] *School of Environmental Science and Engineering, Shaanxi University of Science and*
*Technology, Xi'an 710021, China*
[b] *Yale–NUIST Center on Atmospheric Environment, International Joint Laboratory on Climate*
*and Environment Change (ILCEC), Nanjing University of Information Science and Technology,*
*Nanjing 210044, China*
[c] *Key Laboratory of Coastal Environmental Processes and Ecological Remediation, Yantai*
*Institute of Coastal Zone Research, Chinese Academy of Sciences, Yantai, 264003, China*
[*]Corresponding Author at: Ningliu Road 219, Nanjing 210044, China.
*E-mail address:* dryanlinzhang@outlook.com or zhangyanlin@nuist.edu.cn (Yan-Lin Zhang).



**Abstract:** Environmentally persistent free radicals (EPFRs) are a new type of substance with potential health risks. EPFRs are widely present in atmospheric particulates, but there is a limited understanding of the size-resolved health risks of these radicals. This study first reported the exposure risks and source of EPFRs in atmospheric particulate matter (PM) of different particle sizes (<10 μm) in Linfen, a typical coal-burning city in China. The type of EPFRs in fine particles (< 2.1 μm) is different from that in coarse particles (2.1-10 μm) in both winter and summer. However, the EPFR concentration is higher in coarse particles than in fine particles in summer, and the opposite trend is found in winter. In both seasons, combustion sources are the main sources of EPFRs with coal combustion as the major contributor in winter, while biomass combustion is the major source in summer. Dust contributes part of the EPFRs and it is mainly present in coarse particles in winter and the opposite in summer. The upper respiratory tract was found to be the area with the highest risk of exposure to EPFRs of the studied aerosols, with an exposure equivalent to that of approximately 21 cigarettes per person per day. Alveolar exposure to EPFRs is equivalent to 8 cigarettes per person per day, with combustion sources contributing the most to EPFRs in the alveoli. This study helps us to better understand the potential health risks of atmospheric PM with different particle sizes.

**Key words:** EPFRs; particle size distribution; source; generation mechanism





34 **TOC Art:**

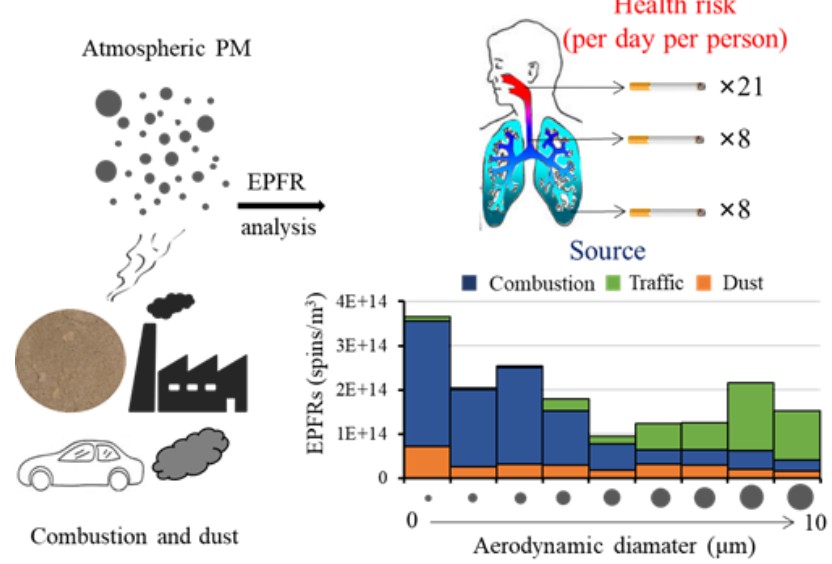

35



## 1. Introduction

Free radicals are atoms or groups containing unpaired electrons, such as hydroxyl radicals and superoxide radicals, and they usually have strong chemical reactivity and short lifetimes (Pryor et al., 1986; Finkelstein., 1982). Free radicals with long lifetimes (months or even years) in the environment are currently called environmentally persistent free radicals (EPFRs), which have received much attention in recent years as new environmentally hazardous substances (Vejerano et al., 2018; Gehling, 2013; Chen et al., 2019c). EPFRs can be used as an active intermediate to catalyze the production of reactive oxygen species (ROS) by oxygen molecules, thus endangering human health (D'Arienzo et al., 2017; Thevenot et al., 2013; Harmon et al., 2018; Blakley et al., 2001; Khachatryan et al., 2011). Studies have found that EPFRs are present in different environmental media, such as water and soil, and even in the atmosphere (Dellinger et al., 2001; Truong et al., 2010; Vejerano et al., 2012).

A number of studies have investigated the occurrences, sources and formation mechanisms of EPFRs in atmospheric particulates in different regions. For example, in the studies of Rostock in Germany, Taif in Saudi Arabia and Xuanwei in China, the average concentration of EPFRs in atmospheric particulate matter (PM) was reported to be in the range of $\sim 10^{16}$ - $10^{18}$ spins/g (Wang et al., 2018; Arangio et al., 2016; Shaltout et al., 2015). Atmospheric EPFRs are mainly carbon-centered radicals with adjacent oxygen atoms (Gehling et al., 2013). EPFRs of different lifetimes are present in atmospheric PM, with only a few hours for short-lifetime EPFRs and several years for long-lifetime EPFRs that show no signs of decay (Gehling et al., 2013; Chen et al., 2019c). Most studies indicate that sources of transportation and combustion may be the primary EPFR sources in atmospheric PM (Wang et al., 2018; Yang et al., 2017; Chen et al., 2019b). Chen et al. (2018b and 2019b) found that strong atmospheric photochemical effects in summer and dust particles may also be important sources of EPFRs. The process of electron transfer and stabilization between the surface of metal oxides (such as iron, copper, zinc and nickel) and substituted aromatic molecules





under high temperatures is considered to be the main mechanism for the formation of
EPFRs in atmospheric particles (Truong., 2010; Vejerano et al., 2012a; Patterson et al.,
2013; Vejerano., 2010; Vejerano et al., 2012b). However, the study by Chen et al.
(2018a) suggests that EPFRs in atmospheric particulates are mainly derived from
graphite oxide-like substances produced during combustion. In addition to primary
sources such as combustion, secondary chemical processes in the atmosphere may
also be an important source of EPFRs in atmospheric PM (Chen et al. 2019b and
2019d; Tong et al., 2018).
Different particle sizes of atmospheric PM pose different health risks to humans,
depending on the deposition efficiency of the particles and the chemical composition
and concentrations of hazardous substances they contain (Strak et al., 2012;
Valavanidis et al., 2008). Among various hazardous substances, EPFRs may also be
involved in the toxicity of atmospheric particulates. Yang et al. (2017) studied the the
EPFRs that are extractable by dichloromethane in different particle sizes in Beijing in
winter and found that the concentration of EPFRs was the highest in particles with
sizes < 1 μm. Arangio et al. (2016) found that the concentration of EPFRs in 180 nm
particles was the highest in the 56 nm - 1.8 μm particle size range. Although several
studies have examined the particle size distribution of EPFRs, systematic studies have
not been conducted on the formation mechanism, source and exposure assessment of
EPFRs in atmospheric particles with different particle sizes.
This study takes Linfen as an example. Linfen is one of the cities in China with
the most serious air pollution and is a typical coal-burning city. The particle size
distribution of EPFRs in atmospheric PM in this region was studied by EPR
spectrometry. The effects of particle size and season on the source, formation
mechanism, and health risk of EPFRs were revealed. In particular, the comprehensive
health risks of EPFRs were evaluated, and it was found that the upper respiratory tract
is the area with the highest risk of EPFRs exposure, which is equivalent to twenty-one
cigarettes per person per day. This study is of great significance for understanding the
source and formation mechanism of EPFRs in atmospheric particulates as well as for





health risk assessments.

## 2. Experimental section

*2.1 Sample collection*

The sampling site for this study is located in Hongdong (36°23', 111°40'E) in
Shanxi, China. To collect atmospheric particles of different sizes (0-10 μm), this study
used a Thermo-Anderson Mark II sampler to collect aerosol samples of 9 sizes. The
samples were collected on a prebaked quartz filter (450 °C, 4.5 hours), and the
sampling dates were as follows: in winter, January 26 to February 4, 2017, $n = 10$; and
in summer, July 31 to August 24, 2017, $n = 12$. The samples were placed in a -20 °C
refrigerator prior to analysis.

*2.2 EPFR analysis*

Specific testing protocols have been described previously (Chen et al., 2018c. The
sample filter was cut into thin strips and clamped with a quartz piece, and then the
quartz piece with attached filter sample was placed in a resonant cavity and analyzed
by an EPR spectrometer (MS5000, Freiberg, Germany). The detection parameters
were magnetic field strength, 335 - 342 mT; detection time, 60 s; modulation
amplitude, 0.20 mT; number of detections, 1; and microwave intensity, 8.0 mW.

*2.3 Carbon composition analysis*

The contents of organic carbon (OC) and elemental carbon (EC) in the filter
samples were analyzed using a semicontinuous OC/EC analyzer (Model 4, Sunset Lab.
Inc., Oregon, USA) with a NIOSH 5040 detection protocol (Lin et al., 2009).
The water-soluble organic carbon (WSOC) concentration was analyzed using an
automatic TOC-LCPH analyzer (Shimadzu, Japan). The WSOC extraction was
performed with ultrapure water under ultrasonication for 15 minutes, and all WSOC
concentrations were blank corrected. The concentration of OC in the MSM
(Methanol-soluble materials) was calculated as the difference between the OC and





WSOC (Water-soluble organic carbon) concentrations. This calculation assumes that
all water-insoluble organic carbon (WISOC) in the aerosol can be extracted with
MeOH, and the rationality of this assumption has been verified elsewhere (Mihara et
al., 2011; Liu et al., 2013; Cheng et al., 2016; Chen et al., 2019a).
*2.4 PAH analysis*
PAHs were detected using gas chromatography/mass spectrometry (GC/MS) on a
GC7890B/MS5977A (Agilent Technologies, Clara, CA), as described in detail in a
previously published study (Han et al., 2018).
*2.5 Metal element analysis*
The concentration of metal elements in the samples was determined by a Thermo
X2 series inductively coupled plasma mass spectrometer (ICP-MS, Thermo, USA).
The metal elements analyzed in summer were Na, Mg, K, Ca, Ti, V, Cr, Mn, Fe, Co,
Ni, Cu, Zn, As, Cd, Pb, and Al, and those in winter were Al, Zn, V, Cr, Mn, Co, Ni, Cu,
As, Se, Sr, Cd, Ba, and Pb. The specific measurement method is based on the study of
Qi et al (2016).
*2.6. Data statistics method*
The source and formation mechanism of EPFRs in PM with different particle sizes
were analyzed by nonnegative matrix factorization (NMF). The method is based on
the study of Chen et al (2016 and 2019e). Briefly, NMF analysis of EPFR data, metal
element contents, OC/EC contents and PAH contents was performed in MATLAB.
The version of the NMF toolbox is 1.4 (https://sites.google.com/site/nmftool/). First, a
gradient-based multiplication algorithm is used to find a solution from multiple
random starting values, and then the first algorithm is used to find a solution to the
final solution using a least squares effective set algorithm. To find a global solution,
the model was run 100 times, each time with a different initial value. By comparing
the 1-12 factor model (Figure S4) with the residual of the spectral load, the 6 factor
(summer) and 10 factor (winter) NMF models were finally selected.




*2.7. EPFR exposure evaluation*
To assess the health risks of EPFRs, we divided the respiratory system into three
parts based on the human breathing model: extrathoracic (ET) areas, including the
anterior nasal cavity, posterior nasal cavity, oral cavity, and throat; tracheobronchial
(TB) areas, including the trachea, bronchi, bronchioles, and terminal bronchi; and
pulmonary (P) areas, including the alveolar ducts and alveoli. Then, the sedimentation
rates of different particle sizes in different areas of the respiratory system were
determined to calculate the exposure risk of EPFRs. Here, the human respiratory
system particulate deposition model of Salma et al. (2002) was used, and the specific
data can be found in Table S3 and S4.
In addition, we converted the daily inhaled concentration of EPFRs into the
concentration of free radicals in cigarettes. The specific conversion method is as
follows:
$$N_{cig} = (C_{EPFRs} \cdot V)/(RC_{cig} \cdot C_{tar}) \tag{1}$$
where $N_{cig}$ represents the number of cigarettes (/person/day), $C_{EPFRs}$ (spins/m$^3$)
represents the atmospheric concentration of EPFRs in PM, and V represents the
amount of air inhaled by an adult per day (20 m$^3$/day) (Environmental Protection
Agency, 1988). $RC_{cig}$ (4.75 × 10$^{16}$ spins/g) (Baum et al., 2003; Blakley et al., 2001;
Pryor et al., 1983; Valavanidis and Haralambous, 2001) indicates the concentration of
free radicals in cigarette tar, and $C_{tar}$ (0.013 g/cig) indicates the amount of tar per
cigarette (Gehling et al., 2013).

## 3. Results and discussion

*3.1 Concentrations and types of EPFRs*
Figure 1a shows the concentration distribution of EPFRs with different particle
sizes in different seasons. EPFRs were detected in the particles of each tested size (the
EPR spectrum is shown in Figure S1), but their EPFR concentration levels were
different. In summer, the concentration of EPFRs in fine particles (particle size < 2.1



μm) is $(3.2 - 8.1) \times 10^{13}$ spins/m$^3$, while the concentration of EPFRs in coarse
particles (particle size > 2.1 μm) is 1-2 orders of magnitude higher than that of fine
particles, reaching values of $(2.2 - 3.5) \times 10^{14}$ spins/m$^3$. Winter samples show
completely different characteristics from summer samples. The concentration of
EPFRs in fine particles (particle size < 2.1 μm) is $(1.8 - 3.6) \times 10^{14}$ spins/m$^3$, while the
concentration of EPFRs in coarse particles (particle size > 2.1 μm) is smaller than that
of fine particles, with values of $(1.0 - 2.1) \times 10^{14}$ spins/m$^3$. In addition, the
concentration of EPFRs in particulates <0.43 μm in winter is very high, but it is very
low in summer. This particulate matter is related to combustion, which indicates that
coal combustion in winter may provide an important contribution to EPFRs. The
EPFR concentration in the fine PM of Linfen reported above is equivalent to that in
the fine PM of Xi'an, but it is ten times smaller than that in the fine PM of Beijing
(Yang et al., 2017; Chen et al., 2019b). Although the particle size distribution
characteristics of EPFRs in winter and summer are different, their concentration levels
are similar, which indicates that the EPFR concentration is not related to the PM
concentration, but is determined by the source characteristics. The source
characteristics will be discussed in detail in the factor analysis section.

Figure 1b shows the contribution of the EPFR concentration to the overall EPFR

concentration in coarse and fine particles. The contribution of fine PM in summer is
only 14.9%, while that of fine PM in winter is 58.5%. The differences in EPFR
concentrations with particle size may be related to the source of EPFRs. For example,
coarse particles are often associated with dust sources. In another study, we have
shown that dust particles contain large amounts of EPFRs and that they can be
transported over long distances (Chen et al., 2018b). EPFRs in fine particles may be
mainly derived from the combustion process, such as traffic sources, which are
considered to be an important source of EPFRs in atmospheric PM (Secrest et al.,
2016; Chen et al., 2019b). Due to winter heating in the Linfen area, the amount of
coal burning increases sharply in this season. In 2017, the nonclean heating
(Coal-fired heating) rate of urban heating energy structures in Linfen was 40% (data



source: http://www.linfen.gov.cn/). With the burning of coal, large amounts of EPFRs
are produced, and in the summer, EPFRs emitted by burning coal should be much less
than those emitted in winter. This can explain to a certain extent that the contribution
of fine particles to summer EPFRs is small, and the contribution of winter EPFRs is
very large.

The $g$-factor is a parameter used to distinguish the type of EPFR (Shaltout et al.,

2015; Arangio et al., 2016). The $g$-factor of carbon-centered persistent free radicals is
generally less than 2.003, the g factor of oxygen-centered persistent radicals is
generally greater than 2.004, and the g factor of carbon-centered radicals with
adjacent oxygen atoms is between 2.003 and 2.004 (Cruz et al., 2012). Figure 2a
shows the $g$-factor distribution characteristics of EPFRs in different particle sizes in
summer and winter. The $g$-factor of fine particles and coarse particles also shows
different characteristics. The $g$-factor of EPFRs in fine particles (particle size < 2.1
μm) ranges from 2.0034 to 2.0037, which may be from carbon-centered radicals with
adjacent oxygen atoms. However, the $g$-factor of EPFRs in coarse particles (particle
size > 2.1 μm) is significantly less than that of fine particles. The $g$-factor ranges from
2.0031 to 2.0033, indicating that EPFRs in coarse particles are more carbon-centered
than those in fine particles and are free of heteroatoms. Although the particle size
characteristics of the $g$-factor of the EPFRs in summer and winter are the same, the
variation in the $g$-factor with concentration is different. As shown in Figure 2b, the
$g$-factor of summer PM showed a significant decreasing trend with increasing
concentration, while the $g$-factor of winter PM showed a significant increasing trend
with increasing EPFR concentration. Oyana et al. (2017) studied EPFRs in the surface
dust of leaves in the Memphis region of the United States and found that the
concentration of EPFRs was positively correlated with the $g$-factor, and they believed
that this was related to the source of EPFRs. This phenomenon indicates that the
sources and toxicity of EPFRs in winter and summer are different. Figure 1 shows
that the summer EPFRs are mainly derived from coarse particles, while the $g$-factor of
EPFRs in coarse particles is smaller than that in fine particles, so the $g$-factor of



EPFRs in summer decreases with an increase in EPFR concentration. In winter, fine
particles contribute more to EPFRs, so the *g*-factor of EPFRs in winter increases with
the concentration of EPFRs.
*3.2 Factor Analysis of EPFRs*
To explore the possible sources and formation mechanism of EPFRs in atmospheric
particles with different particle sizes, the NMF model was used to statistically analyze
EPFRs, carbon components, PAHs and metal elements in samples. The factors
obtained by the NMF model should reflect the different sources mechanisms of
EPFRs. As shown in Figure 3a1 and b1, the three main contributing factors to EPFRs
in summer and winter are shown (see Figure S5, S6 for spectra of other factors),
which explain 94.5% and 83.8% of the EPFR concentrations in summer and winter,
respectively.
As shown in Figure 3a, the typical spectral characteristic of summer factor 1 is that
it contains a small fraction of EC components and a large amount of OC components,
which indicates that combustion may be the source associated with this factor. This
factor has the highest loading of OC, especially WISOC; this fraction mainly contains
macromolecular organic substances, which are considered to contribute to the main
atmospheric particulate EPFRs and to be graphite oxide-like substances (Chen et al.,
2017; Chen et al., 2018a). The result shows that factor 1 has the highest contribution
of all the factors to EPFRs in PM (69.6%), and they are mainly distributed in particles
with sizes > 2.1 μm. Factor 2 is typically characterized by a high contribution from
EC and a small fraction of OC and metal elements, which is a typical source of
incomplete combustion. Factor 2 is different from factor 1; factor 2 is more likely the
combustion of fossil fuels, while factor 1 may be biomass combustion source. The
generation mechanism is similar to a hybrid mechanism, which includes the graphite
oxide-like substances produced by incomplete combustion and the EPFRs formed by
some metal oxides. The relative contribution of these EPFRs is 13.5% and is mainly
distributed in particles with a size < 0.43 μm. The typical characteristic of factor 3 is
that the contribution of metal elements is relatively high, while the contributions of





EC and OC are very low. Metal elements such as Al, Ti, Mn, and Co are typical crust
elements, so this factor may represent dust sources (Pan et al., 2013; Srivastava et al.,
2007; Trapp et al., 2010). The generation mechanism may be mainly due to the
participation of metal oxides in the generation of EPFRs. Compared with the other
factors, this factor also has a partial load on PAHs, indicating that PAHs may be
involved in the formation of metal oxide-related EPFRs. These EPFRs have a
relatively low contribution to total EPFRs (approximately 12.4%) and are mainly
distributed in particles with a size of 0.43 - 2.1 μm. The EPFR contribution of other
factors is 4.4%; they are likely derived from the electroplating metallurgy industry
(detailed in S1).
The results of the factor analysis in winter are different from those in summer. As
shown in Figure 3b, the typical spectral characteristic of factor 1 is that it contains a
large amount of OC components and As and Se. As and Se are trace elements of coal
combustion, as shown in many studies (Pan et al., 2013; Tian et al., 2010), so coal
combustion may be the source represented by this factor. From the generation
mechanism viewpoint, the factor does not contain EC, but the content of OC is very
high. In the particles with a particle size of less than 3.3, which is mainly present in
factor 1, the concentration of OC is 16 times that of EC. So it may be mainly a
graphite oxide-like substance formed by the agglomeration of gaseous volatile organic
compounds (VOCs) generated during combustion. These EPFRs are mainly
distributed in particles with a size of 0.43 - 3.3 μm, and their contribution to EPFRs in
PM is up to 44.6%. Factor 2 contributes 25.7% to EPFRs. The typical spectral
characteristics are due to a large amount of V and some Al, EC and OC. OC and EC
are also typical combustion products. V is rich in fossil fuels, especially fuel oil
(Karnae et al., 2011). Therefore, traffic is the source represented by this factor. The
factor contains crust elements such as Al and Mn, so it is speculated that this factor
may also include traffic-related dust. The particle size distribution shows that such
EPFRs are mainly present in large particles with a size of 3.3 - 10 μm. The typical
spectral characteristics of factor 3 are similar to those of factor 1, and both contain





relatively large amounts of As and Se, with the exception that factor 3 contains a large
amount of EC, indicating that it is also mainly derived from incomplete combustion
sources. The generation mechanism of factor 3 should be different from factor 1,
which may include both the graphite oxide-like material generated by fuel coking and
the EPFRs generated by the metal oxide. These EPFRs are mainly distributed in
particles with a size of <0.43 μm, and their total contribution to EPFRs in PM is
13.4%. In addition, the other factors contribute 16.3% to EPFRs, and these factors are
mainly atmospheric dust (11.4%) and electroplating or metallurgy (4.9%) (see text
S1). The results of this study factor analysis were similar to the results of the study by
Wang et al. (2019) on EPFRs in Xi'an. They found that coal, traffic and dust were the
main sources of EPFRs and accounted for 76.2% of the total source.
Based on the above analysis, it can be found that combustion sources are the main
sources of EPFRs, and EPFRs from these sources are mainly graphite oxide-like
substances generated by the polymerization of organic matter or fuel coking. Studies
have shown that graphene oxide can cause cell damage by generating ROS (Seabra et
al., 2014). The surface of these compounds contains not only carbon atoms but also
some heteroatoms, which leads to disorder and the presence of defects in the
carbon-based structure (Lyu et al., 2018; Chen et al., 2017a; Mukome et al., 2013;
Keiluweit et al., 2010). The dust source is also a source of important EPFRs identified
in this study (with a contribution of approximately 10%). It was shown in the above
analysis that the concentration of EPFRs in coarse particles has a significant
correlation with the concentration of metallic elements, particularly crustal elements.
Some crustal elements, such as Al, and Fe, not only have their own paramagnetism
(Li et al., 2017; Yu et al., 2013; Nikitenko et al., 1992), but also interact with aromatic
compounds attached to the surface of the particles to produce a stable single-electron
structure.
3.3 *Health risk of EPFRs*
To evaluate the health risks of EPFRs in PM with different particle sizes, we





evaluated the comprehensive exposure of EPFRs based on the deposition efficiency of
PM with different particle sizes in different parts of the human body. The results are
shown in Figure 4a. The ET region is the region with the highest EPFR exposure,
while the TB and P regions have relatively close EPFRs. This result shows that
atmospheric EPFRs are the most harmful to the health of the human upper respiratory
tract. Comparing the EPFR exposure in different seasons indicates that the exposure
risk in the ET area in summer is significantly higher than that in winter. This
difference occurs because the concentration of EPFRs in coarse particles is much
higher than that of fine particles in summer and the deposition efficiency of large
particles in the ET area is generally higher. Fine particles are more efficiently
deposited in the P region, leading to a higher risk of EPFR exposure in the P region in
winter.

EPFRs were first found in cigarette tar and are considered one of the health risk

factors in cigarette smoke (Lyons et al., 1960); thus, in this study, the exposure risks
of EPFRs in particles deposited in the human body were converted to the equivalent
number of cigarettes inhaled per adult per day. As shown in Figure 4b, the ET area is
the most contaminated area, with an average equivalence of twenty-one cigarettes
(twenty-five in summer and sixteen in winter). The average values for the TB area
(nine in summer and seven in winter) and P area (seven in summer and ten in winter)
are eight. The results indicate that EPFRs pose significant health risks to human lungs
in both winter and summer. Other similar studies, such as a study of the average
amount of EPFRs in $PM_{2.5}$ inhaled per person per day in Xi'an in 2017, found values
equivalent to approximately 5 cigarettes (Chen et al., 2018a). Gehring et al. (2013)
found that EPFR exposure in $PM_{2.5}$ is equivalent to approximately 0.3 cigarettes per
person per day in St. Joaquin County, the location with the worst air pollution in the
United States. The average exposure risk of EPFRs in fine particles in the Linfen area
(approximately 13 cigarettes) was higher than those in these two studies. However,
these previous studies only studied the exposure risk of EPFRs in fine particles. The
results of this study indicate that the health risks of EPFRs are significantly increased





when the particle size distribution of EPFRs is taken into account. Therefore, it is
important to study the source characteristics and generation mechanism of EPFRs
with different particle sizes, which will be discussed in detail in the following
sections.

This study calculated the proportion of EPFRs with different particle sizes in

different parts of the respiratory system based on the deposition efficiency of particles
with different particle sizes. As shown in Figure 4c, in the ET region and the TB
region, coarse particles are the dominant component in summer and winter. In
particular, in summer, the proportion of EPFRs in coarse particles in these two regions
exceeds 95%. In the P region, there are significant differences between summer and
winter. The P region in summer is still dominated by coarse particles, but its
proportion is significantly lower than those in the ET and TB regions. In the P region
in winter, fine particles are the dominant component (approximately 70%). These
distribution characteristics indicate different sources of EPFRs in different regions. As
shown in Figure 4d, in summer, combustion sources are the main source of EPFRs in
the respiratory system. In winter, combustion and transportation sources contribute
equally in the EP and ET regions, while in the alveoli, combustion sources are the
main contributor. The ET region is the area with the highest risk of exposure to
EPFRs (21 cigarettes). The generation mechanism of these EPFRs is mainly
attributable to graphene oxide-like substances. Studies have shown that graphene
oxide is cytotoxic (Harmon et al., 2018). In the alveoli, the contribution of
combustion sources is significantly increased (especially in winter). These EPFRs are
mainly generated by the action of metal oxides and organic substances. Studies have
shown that such EPFRs can generate ROS in the lung fluid environment (Khachatryan
et al., 2011). Therefore, the health risks of EPFRs from different sources and
mechanisms should be evaluated in the future in order to better assess the harm
caused by EPFRs to the body.
4. **Conclusions and environmental implications**





This study systematically reported the particle size distribution of EPFRs in
atmospheric PM in Linfen, which is one of the most polluted cities in China and is
located in a typical coal-burning area. In addition, this study evaluated the
comprehensive health risks of EPFRs, and reported possible sources and formation
mechanisms of atmospheric EPFRs with respect to different particle sizes. The
following main conclusions were obtained.
(1) This study found that EPFRs are widely present in atmospheric particles of
different particle sizes and exhibit significant particle size distribution characteristics.
EPFR concentrations are higher in coarse particles than in fine particles in summer
and vice versa in winter. Differences were also found in the g-factors of EPFRs in
coarse particles and fine particles, indicating that the types of EPFRs of different
particle sizes were also different. The results of this study demonstrate that the
concentrations and types of EPFRs are dependent on particle size and season. This
result indicates that the potential toxicity caused by EPFRs may also vary with
particle size and season.
(2) This study reported the possible source and formation mechanisms of
atmospheric EPFRs in different particle sizes. The results show that combustion is the
most important source of EPFRs (>70%) in both winter and summer PM samples in
Linfen. Atmospheric dust also contributes to EPFRs (~10%), and they are mainly
found in fine particles in summer and coarse particles in winter. The graphite
oxide-like mechanism has the highest contribution (~70%) and is mainly distributed
in particles with a size of > 0.43 μm, while EPFRs in which metal oxides participate
are mainly distributed in particles with a size of < 0.43 μm. These findings deepen our
understanding of the pollution characteristics of atmospheric EPFRs and are useful for
controlling EPFR generation in heavily polluted areas.
(3) This study assessed the exposure risk of EPFRs in different areas of the
respiratory system. The results show that the upper respiratory tract is the area with
the highest EPFR exposure (the value in summer is higher than that in winter), with a
value equivalent to 21 cigarettes per person per day. EPFRs are equally exposed to the





trachea and alveoli, and the risk of exposure is equivalent to that of 8 cigarettes per
person per day. Coarse particles are the main source of EPFRs in the upper respiratory
tract, while fine particles are mainly involved in the alveoli. In summer, combustion
sources are the main source of EPFRs in various parts of the respiratory system. In
winter, traffic and other combustion sources are the main source of EPFRs in the
upper respiratory tract, and combustion sources mainly contribute to the EPFRs in the
alveoli.
Through this study, we have shown that there are significant differences in the
concentrations and types of EPFRs in particles of different sizes and these differences
are due to the influence of the source and generation mechanism. In the future,
assessments of the particle size distribution and the seasonality of EPFRs in
atmospheric PM should be considered. Health risks are another focus of this study.
We found that the upper respiratory tract is the key exposure area of EPFRs, and the
traffic source is the main source of EPFRs in this area. This finding is significant for a
systematic assessment of the health risks of EPFRs. In view of the complexity and
diversity of the formation mechanisms of EPFRs in actual atmospheric particulates,
the relative contributions of EPFRs generated by different mechanisms and their
associated health risks should be more comprehensively studied in the future.
**Acknowledgments**
This work was supported by the National Natural Science Foundation of China
(grant numbers: 41877354, 41761144056 and 41703102), the Provincial Natural
Science Foundation of Jiangsu grant no. BK20180040), the Natural Science
Foundation of Shaanxi Province, China (2018JM4011) and the fund of Jiangsu
Innovation & Entrepreneurship Team.
**Appendix A. Supplementary data**
Appendix A contains additional details, including the EPR spectra of samples of
different particle sizes, correlations between EPFRs and carbon in particles of
different particle sizes, the results and errors of factor analysis, correlation analysis of



EPFRs with metallic elements, and EPFR exposure in different areas of the human
respiratory tract.
**Code/Data availability:** All data that support the findings of this study are
available in this article and its Supplement or from the corresponding author on
request.
**Author contribution:** Qingcai Chen: Research design, Methodology, Writing -
Original Draft, Writing - Review & Editing, Project administration, Funding
acquisition; Haoyao Sun: Investigation, Sample analysis, Writing - Original Draft,
Writing - Review & Editing, Methodology, Formal analysis; Wenhuai Song:
Investigation, Sample collection, Chemical analysis; Fang Cao: Investigation, Sample
collection; Chongguo Tian: Investigation, Chemical analysis; Yan-Lin Zhang:
Conceptualization, Writing - Review & Editing, Formal analysis, Validation, Funding
acquisition.
**Competing interests:** The authors declare that they have no conflict of interest.

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

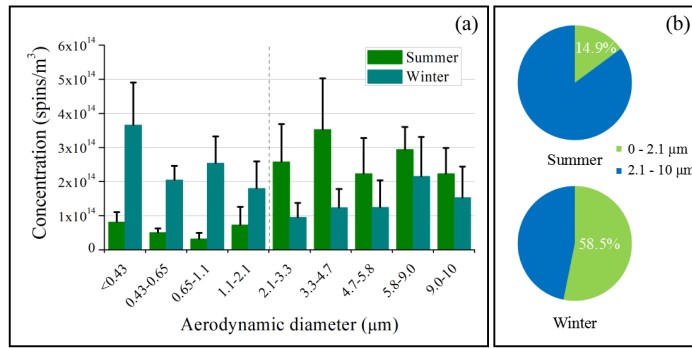


Figure 1. The concentration of EPFRs in PM with different particle sizes. (a) Atmospheric concentrations of EPFRs in different particle sizes in summer and winter. (b) The relative contribution of fine particles and coarse particles to the total EPFR concentration.

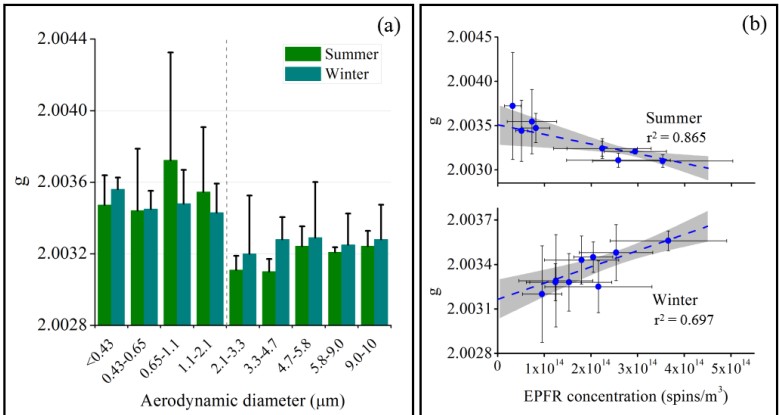


Figure 2. A *g*-factor comparison. (a) Comparison of *g*-factors of EPFRs in different particle sizes

in different seasons. (b) Correlation analysis of *g*-factors and concentrations of EPFRs in summer

and winter PM. The gray areas in the figure represent 95% confidence intervals.



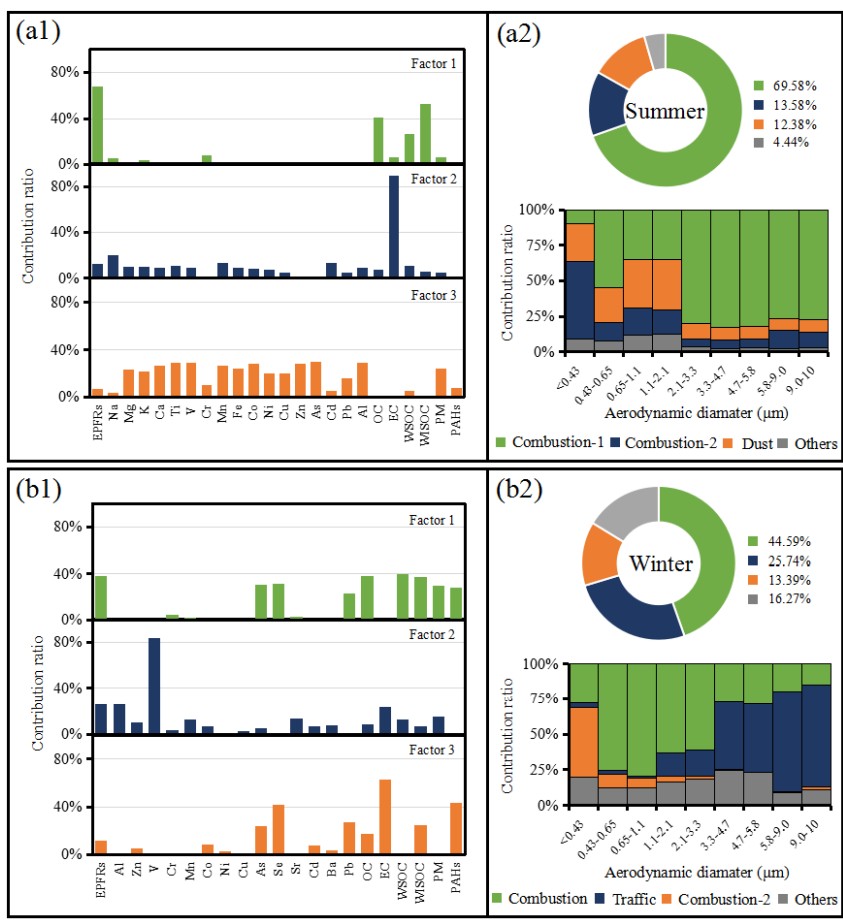

Figure 3. Factor analysis of EPFRs in different particle sizes in different seasons. (a1) and (b1)

represent the results of factor analysis for summer and winter, respectively. (a2) and (b2) represent

the contribution of various factors in summer and winter, respectively, to EPFRs and the relative

contributions of each factor for different particle sizes.

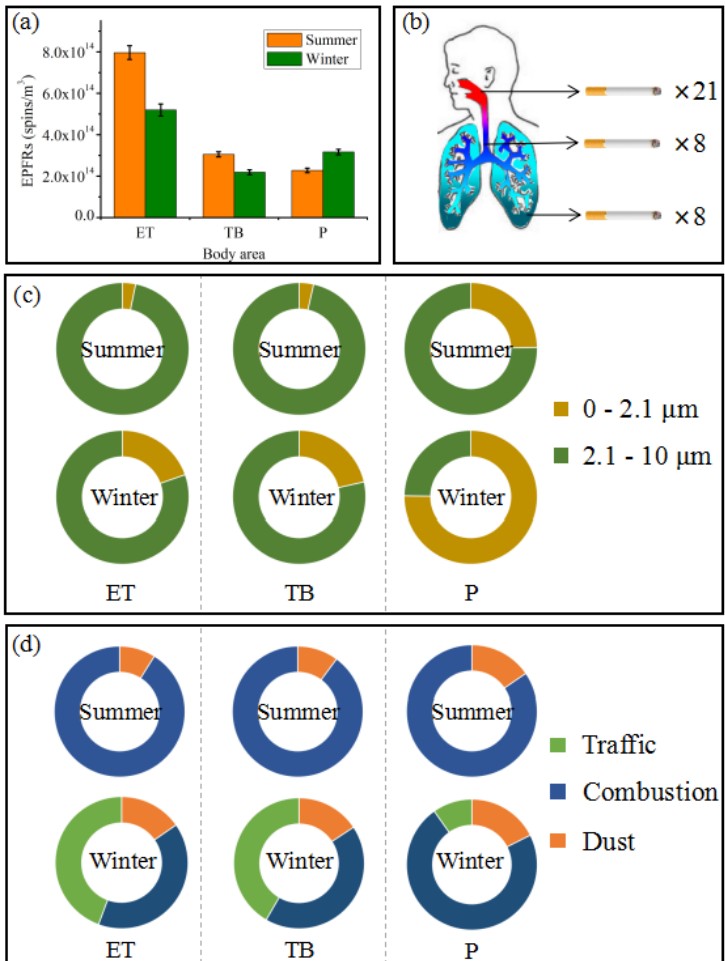

648

Figure 4. Exposure risks to EPFRs. (a) EPFR exposure in the ET, TB, and P regions. (b) Cigarette

exposure to EPFRs in the human respiratory system. (c) Exposure ratio of EPFRs with different

particle sizes in different areas of the respiratory system. (d) Contribution of EPFRs from different

sources to different areas of the respiratory system.