# Peer review of "Size-resolved exposure risk of persistent free radicals (PFRs)"

_Atmospheric Chemistry and Physics, 2020_

## Referee Comment (RC1) · Anonymous Referee #1 · 22 Apr 2020

EPFRs are widely present in atmospheric particulates, but there is a limited understanding of the size-resolved health risks of these radicals. Here, they reported the risks and sources of EPFRs for different particles in summer and winter. They found different types of sources of EPFRs in particles with different sizes. The experimental design was good, and evidence was solid. The results were useful for scientific community, so this paper can be published after the authors address the following comments. Âă It is somehow surprising for me that biomass burning was a major emission source in summer. What was the major types of BB? Open burning? Âă Sec2.2 and 2.4 more details should be given. Âă Lines 139: the link was not active anymore. Âă Line 147: not to use active tense (we or I). Âă Lines 181-182: Âăevidence should be given

to prove it was coal combustion Âă Line 194: it is not necessary dust , and biogenic aerosols can contribute to large particles. Line 207: more details should be given to explain g-factor. Âă I do not suggest using "first" throughout the text. Âă Line 380-389: what can be the implication for such seasonality? what is the driven factor ?

---

## Referee Comment (RC2) · Anonymous Referee #2 · 28 Jul 2020

This paper reports measurements of environmentally persistent free radicals (EPFRs) in particulate matter sampled in Linfen, China. The measurements took place in 2 seasons and involved size-resolved samples. The work makes a contribution to our understanding of this unique group of health actors, so should be published pending the authors addressing some general and specifics comments.

General Comments: There are sections of the Result and Discussion that are repetitious and could be better organized and made more concise. I will point those out in the specific comments, and I strongly recommend the authors go through the paper with an eye towards making it more clear. The authors use the term "formation mechanism"

[Figure]

throughout the paper, but they present nothing that resembles the chemistry that would constitute a formation mechanism. I think the authors need to find a better term that describes what they mean, or show actual chemical mechanisms.

Specific Comments: Line 17: I am always skeptical when people claim 'firsts'. In addition this is phrased in the past tense. Why not just say "This study reports. . ."? Lines 101-102: How long were the samples refrigerated before analysis? Lines 141-142: The phrase "find a solution to the final solution" sounds awkward and should be rephrased. Lines 190-192: The phrasing here is unclear. I think the authors mean the size-segregated contribution of EPFR concentration to the overall. Is this contribution by mass, it's not clear? Line 195: What kind of EPFRs are found in dust particles? Metals? Lines 207-233: This paragraph was hard to follow, I think because the authors skipped around from sentence to sentence in their discussion of g-factor, concentration, size fraction and season. Sometimes a sentence would be referring to the previous sentence, but in a way that was hard to follow. I would like to see this section rearranged so that it has a more logical and clear flow. Pick one feature at a time and make sure it is clear in each sentence what is being referred to. Lines 244: Could it be that the POC in these samples is actually from secondary organic aerosol formation? Lines 254-256: Here the authors are talking about a graphite oxide formation mechanism – this would be greatly improved if they could should the actual chemical reactions – that is what constitutes a mechanism. Lines 243-314: These paragraphs have the problems as the discussion of g-factors. Everything is mixed together, with sentences that are hard to follow. I suggest really trying to reorganize this so that it is easier to follow. Line 339: I believe this should be Gehling and Dellinger, (2013). Lines 402-403: This sentence is backwards, the trachea and alveoli are exposed to EPFRs not the other way around.

Figures: It is hard to distinguish the blue and green colors in the (a) panels of Figures 1 and 2. Please choose better colors.

Supplement In the first paragraph there is superscript 3 – is this supposed to be a

reference? Figure S7 - the caption and axis – 'modle' should be 'model'.

---

## Author Comment (AC1) · 16 Aug 2020

**Response to the comments by the reviewers**
**Qingcai Chen, Haoyao Sun and Yanlin Zhang.**

We appreciate the comments from two reviewers. We have answered the comments of the two reviewers, and addressed the problems raised by the reviewers, such as inaccurate descriptions of the details of this article and logical problems. These improvements have a very positive effect on this article.

Our responses to all the comments from the reviewers and changes made in the paper are listed below.

*Reviewer #1:*

*EPFRs are widely present in atmospheric particulates, but there is a limited understanding of the size-resolved health risks of these radicals. Here, they reported the risks and sources of EPFRs for different particles in summer and winter. They found different types of sources of EPFRs in particles with different sizes. The experimental design was good, and evidence was solid. The results were useful for scientific community, so this paper can be published after the authors address the following comments.*

We appreciate the positive evaluation of this work.

*Specific comments:*

*(1) It is somehow surprising for me that biomass burning was a major emission source in summer. What was the major types of BB? Open burning?*

We appreciate this comment from the reviewer. The results from factor analysis shown that EPFRs mainly from the combustion sources both in winter and summer. However, the results also showed that the dominant factors were different in winter and summer. Obviously the winter is coal burning and thus the summer should be other combustion sources instead of burning coal. According to the production structure of Linfen area, wheat is the main local agricultural crop, and there is often a problem of burning wheat straw in summer, so we speculate that biomass burning may be an important source of EPFRs. We note that this result is speculative. We have modified this part as follows:

P2L22-24: "In both seasons, combustion sources are the main sources of EPFRs with coal combustion as the major contributor in winter, while other fuel combustions are the major source in summer."
P11L258-260: "Factor 2 is different from factor 1; factor 2 is more likely the combustion of fossil fuels, while factor 1 should be other combustion sources instead of burning coal, such as biomass combustion."

*(2) Sec. 2.2 and 2.4 more details should be given.*

We appreciate this comment from the reviewer. We have added more details about the analysis of EPFRs and PAHs in the sections of 2.2 and 2.4 as follows:

L103-111 (Sec. 2.2): *EPFR analysis*

The EPR spectrometer (MS5000, Freiberg, Germany) is used to detect EPFRs in atmospheric samples. Cut the sample filter into thin strips (5 mm × 28 mm), and put it into the sample tank of the quartz tissue cell (the size of the sample tank is 10 mm × 30 mm)., Then the quartz tissue cell with attached filter sample was placed in a resonant cavity and analyzed by an EPR spectrometer. The detection parameters were magnetic field strength, 335 - 342 mT; detection time, 60 s; modulation amplitude, 0.20 mT; number of detections, 1; and microwave intensity, 8.0 mW. Specific testing protocols have been described previously (Chen et al., 2018c).

L125-140 (Sec. 2.3): *PAH analysis*

PAHs were detected using gas chromatography/mass spectrometry (GC/MS) on a GC7890B/MS5977A (Agilent Technologies, Clara, CA). Quartz-fiber filter samples (8 mm in diameter) were cut from each 25-mm quartz-fiber filter substrates used on the ELPI impactor stages using a stainless-steel round punch over a clean glass dish and loaded into the TD glass tube. Next, the TD glass tube was heated to 310 °C at a rate of 12 °C/min and thermally desorbed at 310 °C for 3 min. The desorbed organic compounds were trapped on the head of a GC-column (DB-5MS: 5% diphenyl-95% dimethyl siloxane copolymer stationary phase, 0.25-mm i.d., 30-m length, and 0.25-mm thickness). Sixteen target PAHs were identified based on retention time and qualified ions of the standards, including 16 EPA parent PAHs (p-PAHs). The method detection limits (MDLs) ranged from 0.2 pg/mm2 (Ace) to 0.6 pg/mm2 (Incdp). Naphthalene-D8, acenaphthene-D10, phenanthrene-D10, chrysene-D12, and perylene D12 were used for the analytical recovery check. All compounds were recovered with a desorption recovery percentage of > 90%. Specific testing protocols have been described previously (Han et al., 2018).

**(3) Line 147: not to use active tense (we or I).**

We appreciate this comment from the reviewer. We have modified this part in the whole text as follows:

L161-162: "To assess the health risks of EPFRs, this study divided the …"
L170-171: "In addition, the daily inhaled concentration… were converted."
L208-210: "In another study, the results have shown …"
L317-319: "To evaluate the health risks…, this study evaluated the …"
L397-399: "Through this study, the results have shown...."
L401-403: "It is found that the upper respiratory tract …"

**(4) Lines 181-182: evidence should be given to prove it was coal combustion.**

We appreciate this comment from the reviewer. We have modified this part as follows:

L192-196: In addition, the concentration of EPFRs in particulates <0.43 μm in winter is very high, but it is very low in summer. According to the results of factor analysis in part 3.2 of this study, this particulate matter is related to combustion, which indicates that coal combustion in winter may provide an important contribution to EPFRs.

*(5) Line 194: it is not necessary dust, and biogenic aerosols can contribute to large particles.*

We appreciate this comment from the reviewer. As the reviewer said, biogenic aerosols may be an important contribution to coarse particulate matter, but there is no research on EPFRs in bioaerosols. Previous studies have demonstrated that dust is an important source of EPFRs in atmospheric particulates in these western Regions of China (Chen et al., 2018b and 2019b). So, this study highlights the sources of dust. Thus, as suggested by the reviewer we also add a possible source of bioaerosols as follows:

L207-208: For example, coarse particles are often associated with dust sources and biogenic aerosols.

*(6) Line 207: more details should be given to explain g-factor.*

We appreciate this comment from the reviewer. We have modified this part as follows:

L220-226: The *g*-factor obtained by using EPR to detect the sample is an important parameter to distinguish the type of EPFR. It is the ratio of the electronic magnetic moment to its angular momentum (Shaltout et al., 2015; Arangio et al., 2016). The *g*-factor of carbon-centered persistent free radicals is generally less than 2.003, the *g*-factor of oxygen-centered persistent radicals is generally greater than 2.004, and the g factor of carbon-centered radicals with adjacent oxygen atoms is between 2.003 and 2.004 (Cruz et al., 2012).

*(7) I do not suggest using "first" throughout the text.*

We appreciate this comment from the reviewer. As the reviewer said, it is not rigorous enough to use "first" in the full text. We have modified the relevant description in this article.

L17 and L139: "first" has been removed.

L330: "EPFRs were first found in…" has been replaced by "EPFRs were found early in…"

**(8) Line 380-389: what can be the implication for such seasonality? what is the driven factor ?**

We appreciate this comment from the reviewer. This seasonal characteristic is mainly affected by the source characteristics of EPFRs. For example, in winter, EPFRs are mainly found in fine particles. These EPFRs are not only easier to enter the human body, but also due to the smaller $g$ factor and lower oxidation degree, which means that their reactivity is higher and the harm to the human body is greater. We have added more details about the driven factor and the implication of seasonal characteristic of EPFRs as follows:

L381-383: This seasonal characteristic of EPFRs is mainly affected by the PM sources, this result also indicates that the potential toxicity caused by EPFRs may also vary with particle size and season.

*This paper reports measurements of environmentally persistent free radicals (EPFRs) in particulate matter sampled in Linfen, China. The measurements took place in 2 seasons and involved size-resolved samples. The work makes a contribution to our understanding of this unique group of health actors, so should be published pending the authors addressing some general and specifics comments.*

We appreciate the positive evaluation of this reviewer.

*General Comments:*
*There are sections of the Result and Discussion that are repetitious and could be better organized and made more concise. I will point those out in the specific comments, and I strongly recommend the authors go through the paper with an eye towards making it more clear. The authors use the term "formation mechanism" throughout the paper, but they present nothing that resembles the chemistry that would constitute a formation mechanism. I think the authors need to find a better term that describes what they mean, or show actual chemical mechanisms.*

We appreciate this comment from the reviewer. We have deleted the results and simplified the conclusion part of the article.

**Section 4:**
"This study systematically reported the particle size distribution of EPFRs in atmospheric PM in Linfen, which is one of the most polluted cities in China and is located in a typical coal-burning area. In addition, this study evaluated the comprehensive health risks of EPFRs, and reported possible sources and formation process of atmospheric EPFRs with respect to different particle sizes. The following main conclusions were obtained.

(1) This study found that EPFRs are widely present in atmospheric particles of different particle sizes and exhibit significant particle size distribution characteristics. The results of this study demonstrate that the concentrations and types of EPFRs are dependent on particle size and season. This seasonal characteristic of EPFRs is mainly affected by the PM sources, this result also indicates that the potential toxicity caused by EPFRs may also vary with particle size and season.

(2) This study reported the possible source and formation process of atmospheric EPFRs in different particle sizes. The results show that combustion is the most important source of EPFRs (>70%) in both winter and summer PM samples in Linfen. The graphite oxide-like process has the highest contribution (~70%) and is mainly distributed in particles with a size of > 0.43 μm. These findings deepen our understanding of the pollution characteristics of atmospheric EPFRs and are useful for controlling EPFR generation in heavily polluted areas.

(3) This study assessed the exposure risk of EPFRs in different areas of the respiratory system. The results show that the upper respiratory tract is the area with

the highest EPFR exposure. The trachea and alveoli are also exposed to EPFRs, and the risk of exposure is equivalent to that of 8 cigarettes per person per day. Coarse particles are the main source of EPFRs in the upper respiratory tract, while fine particles are mainly involved in the alveoli.

Through this study, the results have shown that there are significant differences in the concentrations and types of EPFRs in particles of different sizes and these differences are due to the influence of the source and generation process. In the future, assessments of the particle size distribution and the seasonality of EPFRs in atmospheric PM should be considered. Health risks are another focus of this study. It is found that the upper respiratory tract is the key exposure area of EPFRs, and the traffic source is the main source of EPFRs in this area. This finding is significant for a systematic assessment of the health risks of EPFRs. In view of the complexity and diversity of the formation process of EPFRs in actual atmospheric particulates, the relative contributions of EPFRs generated by different process and their associated health risks should be more comprehensively studied in the future."

In addition, the term generation mechanism is used in many places in this article, and we have changed it to the generation process (includes Key words, L351, L345, etc.). In addition, based on the reviewers' specific comments, we have rewritten some logically problematic parts as follows.

*Specific comments:*
*(1) Line 17: I am always skeptical when people claim 'firsts'. In addition this is phrased in the past tense. Why not just say "This study reports. . ."?*

We appreciate this comment from the reviewer. As the reviewer said, it is not rigorous enough to use "first" in the full text. We have modified this part as follows:

L17-19: This study reports the exposure risks and source of EPFRs in atmospheric particulate matter (PM) of different particle sizes (<10 μm) in Linfen, a typical coal-burning city in China.

*(2) Lines 101-102: How long were the samples refrigerated before analysis?*

We appreciate this comment from the reviewer. The samples used in this study have a low temperature storage time of 1 year before testing. Our previous research has shown that the proportion of long-life EPFRs (with a lifetime of 3-5 years) in atmospheric samples exceeds 80% (Chen et al., 2019). In addition, we compared the EPR spectra of the same sample before and after storage for 1 year. The results showed that EPFRs did not change significantly. Therefore, long-term storage will not affect the conclusions of this study.

[Figure]

Figure S1 The average EPR spectrum of samples stored before and after 1 year ago. N=9. Sample date: January 25-27, April 20-22, July 11-13 2017. Original refers to the sample used in this article, 1 year means that these samples are stored for 1 year (Chen et al., 2019).

➢ Chen, Q., Sun, H., Mu, Z., Wang, Y., Li, Y., Zhang, L., Wang, M., Zhang, Z., 2019. Characteristics of environmentally persistent free radicals in PM2.5: Concentrations, species and sources in Xi'an, Northwestern China. Environ. Pollut. 247, 18−26.

*(3) Lines 141-142: The phrase "find a solution to the final solution" sounds awkward and should be rephrased.*

We appreciate this comment from the reviewer. We have added the discussion of g-factor changes and EPFRs decay as follows:

L153-156: Use the gradient-based multiplication algorithm to find a solution from multiple random starting values, and then use the first algorithm to find the final solution based on the least squares effective set algorithm.

*(4) Lines 190-192: The phrasing here is unclear. I think the authors mean the size-segregated contribution of EPFR concentration to the overall. Is this contribution by mass, it's not clear?*

We appreciate this comment from the reviewer. This sentence refers to the contribution of EPFRs in coarse and fine particles to the total concentration of EPFRs. We have changed this part as follows:

L204-206: Figure 1b shows the size-segregated contribution of EPFR concentration to the overall. The contribution of fine PM in summer is only 14.9%, while in winter is 58.5%.

*(5) Line 195: What kind of EPFRs are found in dust particles? Metals?*

We appreciate this comment from the reviewer. Such EPFRs are supposed

mainly as a type of metals. According to our previous research results, EPFRs in sand and dust have no correlation with EC, which may be due to the fact that dust and gravel contain many magnetic materials such as $Cu^{2+}$, $Mn^{2+}$ and $Zn^{2+}$. They are not only paramagnetic, but may also react with some organic matter to form EPFRs and attach to atmospheric particles. We have added the kind of EPFRs in the test as follows:

L209: "…particles contain large amounts of metallic EPFRs and that…"

➤ Chen, Q., Wang, M., Sun, H., Wang, X., Wang, Y., Li, Y., Zhang, L., Mu, Z., 2018b. Enhanced health risks from exposure to environmentally persistent free radicals and the oxidative stress of PM2.5 from asian dust storms in erenhot, Zhangbei and Jinan, China. Environ. Int. 123, 260−268.

***(6) Lines 207-233: This paragraph was hard to follow, I think because the authors skipped around from sentence to sentence in their discussion of g-factor, concentration, size fraction and season. Sometimes a sentence would be referring to the previous sentence, but in a way that was hard to follow. I would like to see this section rearranged so that it has a more logical and clear flow. Pick one feature at a time and make sure it is clear in each sentence what is being referred to.***

We appreciate this comment from the reviewer. There are some problems with the logic of this paragraph, we have rewritten it as follows:

L220-242: The *g*-factor obtained by using EPR to detect the sample is an important parameter to distinguish the type of EPFR. It is the ratio of the electronic magnetic moment to its angular momentum (Shaltout et al., 2015; Arangio et al., 2016). The *g*-factor of carbon-centered persistent free radicals is generally less than 2.003, the *g*-factor of oxygen-centered persistent radicals is generally greater than 2.004, and the g factor of carbon-centered radicals with adjacent oxygen atoms is between 2.003 and 2.004 (Cruz et al., 2012). Figure 2a shows the *g*-factor distribution characteristics of EPFRs in different particle sizes in summer and winter. The *g*-factor of fine particles and coarse particles shows different characteristics. The *g*-factor of EPFRs in fine particles (particle size < 2.1 μm) ranges from 2.0034 to 2.0037, which may be from carbon-centered radicals with adjacent oxygen atoms. However, the *g*-factor of EPFRs in coarse particles (particle size > 2.1 μm) is significantly less than that of fine particles. The *g*-factor ranges from 2.0031 to 2.0033, indicating that EPFRs in coarse particles are more carbon-centered than those in fine particles and are free of heteroatoms. As shown in Figure 2b, the variation in the *g*-factor with concentration in different season is different. The *g*-factor of summer PM showed a significant decreasing trend with increasing concentration, while the *g*-factor of winter PM showed a significant increasing trend with increasing EPFR concentration. Oyana et al. (2017) studied EPFRs in the surface dust of leaves in the Memphis region of the United States and found that the concentration of EPFRs was positively correlated with the *g*-factor, and they believed that this was related to the source of EPFRs. This

phenomenon indicates that the sources and toxicity of EPFRs in winter and summer are different.

**(7) Lines 244: Could it be that the POC in these samples is actually from secondary organic aerosol formation?**

We appreciate this comment from the reviewer. The dominant factor of factor 1 is WISOC, which is typical of a primary combustion source. On the one hand, according to the generation characteristics of EPFRs, the dominant component of the aromatic substances EPFRs produced by low-temperature combustion. On the other hand, according to the local pollution characteristics, summer burning mainly comes from the burning of straw and the catering process. Therefore, we believe that factor one may be mainly biomass combustion. According to previous studies, EPFRs generated by the secondary process are usually active, with a life span of only tens of minutes, so it is unlikely that they are secondary aerosols.

> ➤ Chen, Q., Sun, H., Wang, M., Wang, Y., Zhang, L., Han, Y., 2019. Environmentally persistent free radical (EPFR) formation by visible-light illumination of the organic matter in atmospheric particles. Environ. Sci. Technol, 53 (17), 10053−10061.

**(8) Lines 254-256: Here the authors are talking about a graphite oxide formation mechanism – this would be greatly improved if they could should the actual chemical reactions – that is what constitutes a mechanism.**

We appreciate this comment from the reviewer. The research on the chemical reaction of the generation mechanism of graphene oxide to EPFRs has been carried out in our previous research. The research on the chemical reaction of the generation mechanism of graphene oxide to EPFRs has been carried out in our previous research. In that study, we conducted high-temperature treatment experiments on actual atmospheric samples and glucose, and performed EPR, OC/EC and FT-IR tests on the processed samples. The experimental results show that the processed sample can generate EPFRs and is rich in benzene ring structure (benzene ring C=C) and heteroatom functional groups. Its EPR spectrum and $g$-factor are similar to graphene oxide.

> ➤ Chen, Q., Sun, H., Wang, M., Mu, Z., Wang, Y., Li, Y., Wang, Y., Zhang, L., Zhang, Z., 2018. Dominant fraction of EPFRs from Nonsolvent-Extractable organic matter in fine particulates over Xi'an, China. Environ. Sci. Technol. 52 (17), 9646−9655.

**(9) Lines 243-314: These paragraphs have the problems as the discussion of g-factors. Everything is mixed together, with sentences that are hard to follow. I suggest really trying to reorganize this so that it is easier to follow.**

We appreciate this comment from the reviewer. We have rewritten this part.

[revised manuscript text omitted]

*(10) Line 339: I believe this should be Gehling and Dellinger, (2013).*

We appreciate this comment from the reviewer. We have corrected this mistake.

*(11) Lines 402-403: This sentence is backwards, the trachea and alveoli are exposed to EPFRs not the other way around.*

We appreciate this comment from the reviewer. We have corrected this mistake as follows:

L393-394: The trachea and alveoli are also exposed to EPFRs, and the risk of exposure is equivalent to that of 8 cigarettes per person per day.

*(12) It is hard to distinguish the blue and green colors in the (a) panels of Figures 1 and 2. Please choose better colors.*

We appreciate this comment from the reviewer. We have changed the colors in panel (a) of Figures 1 and 2 to green and red as follows:

[Figure]

Figure 1. The concentration of EPFRs in PM with different particle sizes. (a) Atmospheric concentrations of EPFRs in different particle sizes in summer and winter. (b) The relative contribution of fine particles and coarse particles to the total EPFR concentration.

[Figure]

Figure 2. A *g*-factor comparison. (a) Comparison of *g*-factors of EPFRs in different particle sizes in different seasons. (b) Correlation analysis of *g*-factors and concentrations of EPFRs in summer and winter PM. The gray areas in the figure represent 95% confidence intervals.

*(13) Supplement In the fifirst paragraph there is superscript 3 – is this supposed to be a reference? Figure S7 - the caption and axis – 'modle' should be 'model'.*

We appreciate this comment from the reviewer. The superscript 3 in the supplementary information is an error and has been deleted. We have modified Figure S7 as follows:

[Figure]

**Figure S7.** Comparison of the concentration of modle export and actual measurement.

---

## Author Response (AR2)

**Response to the comments by the reviewers**
**Qingcai Chen, Haoyao Sun and Yanlin Zhang.**

We appreciate the comments from editor and the reviewers. We have answered the supplementary comments and improved the sentence expression of the article. We look forward to the successful publication of this article.

Our responses to the comments from the reviewers and changes made in the paper are listed below.

*The authors have done a good job responding to the referee's comments. There remain a few points where some clarification or correction is needed.*

We appreciate the positive evaluation of this work.

*Specific comments:*
*(1) In their response to Reviewer #1 concerning Section 2.2, the authors say "Cut the sample filter into thin strips". It would be much better if they said "The filters were cut into thin strips".*

We appreciate this comment from the reviewer. We have modified this part as follows:

L105-107: The filters were cut into thin strips (5 mm × 28 mm), and put it into the sample tank of the quartz tissue cell (the size of the sample tank is 10 mm × 30 mm).

*(2) In their response to Reviewer #1 concerning Section 2.3, It is not clear what is meant by "qualified ions".*

We appreciate this comment from the reviewer. Qualified ions here refers to the typical ion fragments of each PAHs, which was used to determine the type of PAHs. We have provided another reference that describes the PAHs analysis in detail (Song et al., 2020). In addition, we also changed the expression about this point as follows:

L134-136: Sixteen target PAHs were identified based on retention time and typical ion fragments of each PAH standards, including 16 EPA parent PAHs (p-PAHs).

L141-142: Specific testing protocols have been described previously (Han et al., 2018; Song et al., 2020).

Song, W., Cao F., Lin Y., Haque, M. M., Wu, X., Zhang, Y., Zhang, C., Xie, F., Zhang Y., 2020. Extremely high abundance of polycyclic aromatic hydrocarbons in aerosols from a typical coal-combustion rural site in China: Size distribution, source identification and cancer risk assessment. Atmos. Res. 248, 105192.

*(3) The response that explains the g-factor: Instead of "using EPR to detect the sample", it would be more correct to say "using EPR to analyze the sample". This phrase appears in several places, so should be corrected everywhere.*

We appreciate this comment from the reviewer. We have modified this section as follows:

L221-222: The *g*-factor obtained by using EPR to analyze the sample is an important parameter to distinguish the type of EPFR.

*(4) The response to the Reviewer's question on L204-206, the authors have still not specified if the percentages were by mass.*

We appreciate this comment from the reviewer. It is not the contribution by PM mass but is the concentration of EPFRs. We have modified this section as follows:

L205-206: Figure 1b shows the concentration ratio of EPFRs in coarse and fine particles. The contribution of EPFRs in fine PM in summer is only 14.9%, while in winter is 58.5%.

*(5) The authors response L220-242. The phrase "the variation in the g-factor with concentration in different season is different." Is a bit awkward, it would be better to simply say "the g-factor varied differently depending on season".*

We appreciate this comment from the reviewer. We have modified this section as follows:

L235-236: As shown in Figure 2b, the g-factor varied differently depending on season.

*(6) The response to Comment (7). What is "the catering process"?*

We appreciate this comment from the reviewer. The catering process refers to the process of cooking, mainly including cooking fume and the process of burning biomass on stoves in rural areas.

*(7) Line 24 , Instead of "other fuel combustions", just say "other fuels"*

We appreciate this comment from the reviewer. We have modified this section as follows:

L24: …while other fuels are the major source in summer.

*(8) Line 159. The phrase that was changed is better, but should be changed to the past tense. "A gradient-based multiplication algorithm was used to find a solution….. and then the first algorithm was used….. based on the least-squares effective-set algorithm."*

We appreciate this comment from the reviewer. We have modified this section as follows:

[revised manuscript text omitted]